# Determinants of Ship Management Revenues: The Case of Cyprus

Nektarios A. Michail [1], Konstantinos D. Melas [2,3,4,*] and Kyriaki G. Louca [1]

1. Economic Analysis and Research Department, Central Bank of Cyprus, Nicosia 1076, Cyprus; nektariosmichail@centralbank.cy (N.A.M.); kyriakilouka@centralbank.cy (K.G.L.)
2. Faculty of Business and Economics, Metropolitan College, 54624 Thessaloniki, Greece
3. Department of Economics, University of Western Macedonia, Fourka Area, 52100 Kastoria, Greece
4. Department of Supply Chain Management, International Hellenic University, 57001 Thessaloniki, Greece
* Correspondence: mkonstantinos@mitropolitiko.edu.gr

**Abstract:** We explore, for the first time in the literature, how the revenues of ship management companies respond to macroeconomic exogenous shocks. Using data for ship-management companies in Cyprus, we find evidence that a demand shock has the largest impact on revenues, exhibiting an almost one-for-one relationship. If the demand shock is permanent, we observe a ceteris paribus permanent effect on revenues. Similarly, this occurs irrespective of the final effect that demand has on the relevant freight rate, proxied via the Baltic dry and tanker (dirty and clean) indices. The BDI and the BDTI indices have a smaller effect on revenues, standing at approximately 0.05% for every 1% shock, while the clean tanker index does not have an effect, most likely due to their fleet composition. In accordance with the literature, we find that a shock in the price of Brent oil increases revenues. Our results bear importance not only for ship management companies per se, but also for countries that are ship management hubs.

**Keywords:** shipping; ship management; freight rates; Bayesian VAR

**JEL Classification:** G11; G12; G13; G20

## 1. Introduction

Shipping is the most important mode of transportation. According to the most recent statistics, approximately 85% of the world's trade is transported, at some point, by water (UNCTAD 2021). Thus, it comes without surprise that the industry as a whole has been, recently, under the spotlight of the academic, (Alexandridis et al. 2018), investor (Geman et al. 2012), and policymaker communities (Michail et al. 2022).

The fact that maritime transportation is an important factor for the world economy is not only proven by the latter fact but recent research has shown shipping when compared to air and road transportation is more significant for countries' economic growth (Park et al. 2019; Pham and Sim 2020).

As demand for seaborne trade has increased significantly in the current decades (Stopford 2013), investment in vessels attracts ambitious investors (Melas and Michail 2022). However, investing in the maritime sector carries a high degree of risk, given that the industry is characterized by a derived demand (Isserlis 1938; Zannetos 1959; Beenstock and Vergottis 1989a, 1989b) and the macro-economic environment as a whole plays a significant role in the cycle of the specific market. Economic factors like GDP (Michail 2020), interest rates (Mohanty et al. 2021), stock markets (Kavussanos and Marcoulis 2000; Melas and Michail 2021), and even exogenous socioeconomic shocks have a huge effect on the shipping market (Michail and Melas 2020b; March et al. 2021).

In what is perhaps one of the biggest differences in the maritime sector, Stopford (Stopford 2013), in his seminal work, claimed that the shipping cycles last half as much

as the normal business cycles suggesting that market conditions can change much more rapidly than in other sectors. In addition, the magnitude of shipping cycles is one of the main factors that many investors enter the market in the first place. There have been various examples of shipping investors who, in a few years, have become enormously successful (LaRocco 2012).

The opportunity for high profits does matter to investors. While the average return of shipping investments is relatively low when compared to other sectors, their standard deviation of the annual return appears to be much higher, as suggested by the higher magnitude of the cycles (Stopford 2013). This means that even though investors would be better off investing in the stock or bond market over the long run, shipping investors could potentially outperform if they time their investments to the shipping cycle. Naturally, while some investors aim to buy vessels at low prices and sell them at high (Moutzouris and Nomikos 2020), this is a rather risky investment and, as most of the researchers suggest, hard to achieve (Alizadeh and Nomikos 2007). In fact, many scholars have characterized such investors as "unicorns" given that, on average, the shipping industry does not produce high returns for its investors over the long run (Stopford 2013).

Nevertheless, the possibility for a "buy low—sell high" scenario gives rise to another play, given that these investors still have assets that can produce a significant amount of income until their sale. Given that most use a high level of gearing in their investments, they can potentially boost their average returns, while at the same time benefit even more during peak periods (Kavussanos and Visvikis 2006). However, most of them are not interested in managing the vessels themselves, and hence they subcontract such operations to ship management companies (Panayides and Gray 2006). In such a scenario, the vessels would produce a rather stable income for their owners, along with some upside potential. As such, the vicissitudes of the shipping cycle would affect them much less.

In the current paper, we examine for the first time in the bibliography the macroeconomic determinants of ship management company revenues. Using a unique dataset for one of the leading ship management hubs in the world, Cyprus, our results show that a demand shock has the largest impact on revenues, exhibiting an almost one-for-one relationship. This occurs irrespective of the final effect that demand has on the relevant freight rate, proxied via the Baltic dry and tanker (dirty and clean) indices. The BDI and the BDTI indices have a smaller effect on revenues while the clean tanker index does not have an effect, most likely due to the ship management companies' fleet composition. In accordance with the literature, we find that a shock in the price of Brent oil increases revenues. Our results bear significant implications for the broader shipping community as they provide evidence of the importance of ship management and its reliance on global macroeconomic conditions.

Following this introduction, the remainder of this paper is organized as follows: Section 2 provides a review of the (scarce) literature related to our analysis, Section 3 offers an overview of the ship management industry, Section 4 describes the methodology and the data used, Section 5 discusses the empirical results obtained, and Section 6 offers conclusions.

## 2. Literature Review

The literature concerning the ship management companies is rather limited and mainly focuses on managerial aspects. The first study on the matter was by Sletmo (1989), who looked into the fact that traditional shipping powers (such as Great Britain or Greece) have been losing part of their national tonnage due to the early stages of the globalization of the maritime industry. Since Sletmo's (1989) research, ship management companies had increased drastically. The reasons for this phenomenon relate to the oil industry majors who took advantage of the availability of tax breaks on ship investment and made capital investments by purchasing vessels during the 1960s, as well as the low freight rates and the devalued sale and purchase market of the early 1970s.

It was not until 2006 that researchers (Panayides and Gray 2006) looked into the industry again. According to Panayides and Gray (2006), while shipowners are in charge of all the operations of their shipping companies, ship management companies are focusing on the basic ship operation and crewing, chartering, sale and purchase, insurance, new building supervision, and claims handling. The growth of services may be attributed to the responsiveness of ship management companies to the needs of the marketplace, a theme we elaborate more on in the following section.

Moreover, Panayides and Gray (2006) focused on the marketing perspective of the ship management companies at the time. More precisely, in their research, they conclude that ship management companies that build long-term client relationships will ensure client retention, reduce transaction costs, and achieve differentiation and competitiveness. Other micro-level studies of ship management companies (Mitroussi 2007, 2013) provide evidence that ship owners more often than not outsource the crewing and the technical management of their vessels to ship management companies. Moreover, ship owners who employed such services were doing so primarily for flexibility and to relieve themselves from economic pressures, as offered to them by management enterprises. In her follow-up research, Mitroussi (2013) focuses on the importance of shipping economic sustainability under the new environmental legislation, and how ship managers can ameliorate the problems that could possibly arise for the ship owners.

As ship management companies increased, the literature has looked into the strategies that some companies follow. In particular, high-performance companies seem to be achieving economies of scale, differentiation (in particular through a wider range of services offered), and market-focus and competitor analysis (Panayides 2010), while the investment in their human capital is of prime importance (Panayides and Gray 2010; Goulielmos et al. 2011).

Some of the latest research is focusing both on the environmental and digitalization fronts when it comes to ship management companies. Poulsen and Sornn-Friese (2015) were the first to look at the energy efficiency that third-party ship management companies are implementing. Their findings suggest that ship managers are generally indifferent in putting into effect energy-efficient strategies if the ship owners do not push them in such a direction. When it comes to digitalization, Mohamed Ali Awadh Timimi (2021) provided evidence that ship management companies are not providing enough resources for this new era in the maritime industry.

Overall, while the bibliography has given some evidence on how ship management companies operate, no research is available on the macroeconomic factors that can potentially affect them. In addition, there is no overview of the industry as a whole, and the origins and the factors that have mostly affected ship management are usually not presented clearly. To address this gap, the following section offers an introduction to how the ship management industry came to be, while also presenting the factors that have made Cyprus one of the most important shipping hubs in the world.

## 3. The Ship Management Industry and Cyprus

As explained in the previous section, ship management is very important for the maritime industry as a whole, and more specifically for shipping investors who wish to take a more passive role. It is thus not surprising that ship management is one of the services that have been around since the very beginning of shipping. According to Sletmo (1989), the rise of ship management companies started after World War II, with the United Nations Conference on Trade and Development (UNCTAD) created in order to promote the interests of developing countries in world trade. In effect, the aim was to boost international trade with developing countries (especially of manufactured goods) and stabilize prices (e.g., via the International Sugar Agreement in 1968) (Oatley 2019).

To further enhance trade, developing countries introduced what came to be known as "flags of convenience". This was done in order to incentive ship owners to register their vessels in countries that had lower taxes than the ones they had been paying until then. This would benefit developing countries by increasing their incomes, which they could

then tunnel to domestic causes. While this practice dates from long, the establishment of the Liberian open registry in 1948 set the example. In this case, 25% of the revenues would go to the Liberian government with another 10% going to fund social programs in Liberia. The remainder was held by the corporation managing the register. Following the registration of its first vessel in 1949, Liberia became the number one registry in the world (DeSombre 2006; Ojala and Tenold 2017).

The "flagging-out" doctrine assisted in the easy establishment of shipping investments in jurisdictions with lower taxation and was highly successful. By the mid-1980s, 25% of the global tonnage was running under the flags of convenience, with the figure rising to more than 50% by 2008 (UNCTAD 2019). As some of these countries provided attractive tax benefits, shipping investors and firms outsourced their vessels to ship management companies that operated from those jurisdictions. This was particularly important in the late 1970s and 1980s, when the increase in freight costs pushed profit margins down. As such, in order to find alternative ways to remain profitable, companies transferred the management of a ship to a company located in another country in order to benefit from lower tax rates (Michail 2018).

This, however, was not the only reason. As Sletmo (1989) states, the easy access to financing created incentives for more investment in the industry, notably also by outsiders. This meant that shipping investors could be less actively involved and still benefit from the proceeds their vessel would bring in. As such, ship owners could passively reap the benefits from the vessel operation, without any knowledge of the industry and without any hassle, until the time they chose to sell it. As such, ship management companies added to their traditional services a bouquet of new operations such as chartering, sale and purchase, insurance, newbuilding supervision, and claims handling to name but a few.

In Cyprus, taxation played a more important role in the development of the ship management industry than finance. In particular, in the late 1970s and early 1980s, German ship-owning firms, aiming to relieve themselves from the high taxation and the strict employment quotas in their home country, found that their vessels could be considered as residents of another country if they paid a one-off fee to the ship manager (Michail 2018). Hence, the country's tax rate of 4.25%, at the time, and the high level of services provided, a result of a long experience in shipping, provided an attractive alternative. Hence, local and foreign firms (mostly ship owners aiming to reduce their tax burden) started setting up ship management firms on the island, making Cyprus one of the pioneers in the ship management industry (Michail 2018). By the 1990s ship management firms started expanding in ship owning, responding to the increased demand for international sea transport which was driven by large political changes such as the opening of Asia and the fall of the Soviet Union, aided by the containerization momentum.

Shipping in Cyprus got another boost in 2004 when the island joined the European Union. As a result of the long negotiations (ranging from 1996 to 2002), the chapter for sea transport was finalized and Cyprus became the only officially approved country within the EU to maintain an open registry. This further increased the types of shipping activities on the island, as the variety of ship types which are allowed to be registered in Cyprus was very wide (Michail 2018).

In a further innovation to the maritime industry, which also provided a significant boost to the ship management sector, Cyprus introduced the tonnage tax law in March 2010. In contrast to the usual corporate tax, the tonnage tax is simply a fixed percentage of the total carrying capacity of a vessel. For example, a vessel that can carry 70,000 DWT pays a larger amount of tax compared to an equivalent vessel with a capacity of 50,000 DWT. Given that the tax is paid on a ship's carrying capacity, this means that the same amount of taxation will be paid each year for the remaining life of the vessel. Thus, a certainty in expenses is ensured, meaning that the firm or the investor does not have to worry about changes in taxation in boom or bust periods. Furthermore, this taxation scheme is available for a wide range of vessels and, most importantly, companies in the shipping industry

can make use of the tonnage tax, even if they are not the ship owners, as long as they are managing it (Michail 2018).

As a result, of the above innovations, the number of ship management firms on the island increased substantially since the late 1970s. The Cypriot economy benefited hugely from this, with more than 20% of the world's third-party management fleet managed by companies based in Cyprus. The overall revenues from the sector average amount to around 4.6% of GDP, with more than 200 companies offering services. Given the importance of the sector for growth, it is thus quite important to be able to assess the revenue that will be coming into the country (Shipping Deputy Ministry of Cyprus 2021).

To elaborate on these factors, in this study, as we estimate, for the first time in the literature, how the shipping markets (namely the dry bulk, dirty tanker, and clean tanker segments) affect the revenues of these companies. Such information is of prime importance not only for the companies but also for the countries that rely on the industry. In the next section, we offer the data and the methods used for the estimation.

## 4. Data and Methods

Consider a Vector Auto-Regression (VAR) model in which $y_{i,t}$ denotes a matrix with $i$ variables relevant to the ship management. The VAR representation is

$$\Delta y_t = \alpha + \sum_{j=1}^{k} \sum \beta_j \Delta y_{t-j} + \varepsilon_t, \ \varepsilon_t \sim N(0, \Sigma) \tag{1}$$

where $y_t$ is a vector of endogenous variables, $\Delta$ is the first difference operator, $j$ is the appropriate lag length and $\varepsilon_t$ denotes the vector of serially and mutually uncorrelated structural innovations, with variance-covariance matrix $\Sigma$. $\beta_j$ are the appropriate coefficients related to lag $j$ of the vector of dependent variables.

In particular, vector $y_t$, in addition to ship management revenues, also includes the main macroeconomic variables, i.e., oil prices and the stock market capitalization index, both of which are expected to have a positive effect on ship management revenues. In particular, oil prices usually tend to have a positive impact on freight rates (Shi et al. 2013; El-Masry et al. 2010; Gavriilidis et al. 2018) as these represent the main vessels expenses, and are usually passed on to the end client. In this context, we expect oil prices to have a positive effect on ship management revenue as well. Similarly, stock market changes tend to have a positive impact on freight rates (Drobetz et al. 2010; Papapostolou et al. 2016), as they proxy for the prevailing macroeconomic environment. Given that shipping is a derived demand system, the better the macroeconomic situation, the more demand for transportation there will be, and hence freight rates will rise. We note here that if the revenue structure is fixed and does not vary over the shipping cycle, then the change in these variables may not affect them.

In addition, to the variables mentioned above, we also include the Baltic Dry Index (BDI), the Baltic Dirty Tanker Index (BDTI), and the Baltic Clean Tanker Index (BCTI) which are employed to capture the behavior of freight rates, i.e., the equilibrium price as per standard theory (Stopford 2013). The adoption of these particular indices is justified by the fact that they represent the vast majority of ocean cargo transported, and are the underlying assets of shipping freight option contracts (Tsouknidis 2016).

Finally, we also use the interest rate in our estimation, for the first time in the literature, in order to account for the potential spillovers from monetary policy and financing conditions. As is well known, capital costs can account for a large part of the total expenses for a vessel, and hence an increase in the financing cost can have important implications for shipping companies. Most importantly, higher policy rates suggest that the overall conditions in an economy are tighter, meaning that demand should be lower. As such, and according to standard economic theory, the interest rate is expected to have a negative effect on revenues. In our model, we proxy the global policy rates via the US Effective Federal Funds Rate (EFFR), given that the US is the largest importer in the world (UNCTAD 2021).



Moving to the particularities of our estimation, as it is widely known, the sample size is essential in this formulation since estimations of $\beta_j$ can be inaccurate when the time-series dimension is small (Weale and Wieladek 2016). This is important given that the data range for our estimation is constrained: while freight indices are available starting from 2000, the ship management revenues data from the Central Bank of Cyprus survey are only available from 2009 onwards. Subsequently, the application of Bayesian methods, as presented by Litterman (Litterman 1986), is used to address this issue. In particular, and as previously introduced in the shipping literature (Michail and Melas 2020a), we use a non-informative normal–inverse Wishart, allowing us to obtain more robust results. The use of this prior is also popular in the economics literature (Uhlig 2005; Weale and Wieladek 2016). In addition, for the VAR specifications, standard hyperparameter values have been used, i.e., a 0.8 auto-regressive coefficient, tightness of 0.1, cross variable weighting of 0.5, and lag decay of 1 and 100 for the exogenous variable tightness. The variables have a lag length of 1[1] and follow a Cholesky identification order.

To avoid the use of the imposition of a Kronecker structure on the prior distribution, which creates a dependence between the variance of the residual term and the variance of the VAR coefficients for each equation, (Dieppe et al. 2016), we use an Independent Normal-Wishart (INW) prior with unknown $\Sigma$ and an arbitrary variance–covariance matrix, $\Omega_0$. Hence, the prior distribution is specified such that, $\beta \sim N(\beta_0, \Omega_0)$. While any structure can be adopted for $\beta_0$ and $\Omega_0$, the former is typically defined as the usual Minnesota $\beta_0$ vector, with one in the first lag of each endogenous variable and zero for further lags and cross-variable lag coefficients (Dieppe et al. 2016). Similarly, $\Omega_0$ also takes the form of the Minnesota covariance matrix. Given these conditional distributions, it is possible to use the Gibbs sampler to obtain random draws from the unconditional posterior distributions of the parameters of interest.

With regards to data sources, data for Brent crude oil prices and the Wilshire 5000 total market full cap index, as well as for the US Effective Federal Funds Rate (EFFR) were collected from Federal Reserve Economic Database (FRED). Data for the freight indices were obtained from Clarksons Shipping Intelligence, while the ship management revenues data are gathered from the Ship Management Survey, conducted by the Statistics Department of the Central Bank of Cyprus (CBC) and concentrates primarily on transactions between resident ship management companies and ship owning/shipping related entities.[2] Our data range from 2009q1 to 2022q2 (full data availability of the survey).

Before we proceed with the estimation, it is useful to illustrate the path of the main variable of interest. Figure 1 shows the path of ship management revenues in the country. As the path shows, revenues have been increasing over time, evidenced by the dashed trend line. Excluding the pandemic period, revenues have almost doubled since 2009, while they have fully recovered to their pre-Covid levels by 2022q2. To illustrate how important these revenues are for the country, we note that they have averaged around 4.6% of Cyprus' GDP.

With regards to the estimation, we note that one lag was used, as this resulted in the lowest log-likelihood value. The BVAR abides by good statistical practices as no roots lie outside the unit circle. To account for the one-off drop during the pandemic, a dummy variable took the value of 1 over the 2020q1–2020q4 period and zero otherwise to account for the one-off COVID-19 pandemic era. Results from our estimation can be found in the following section.

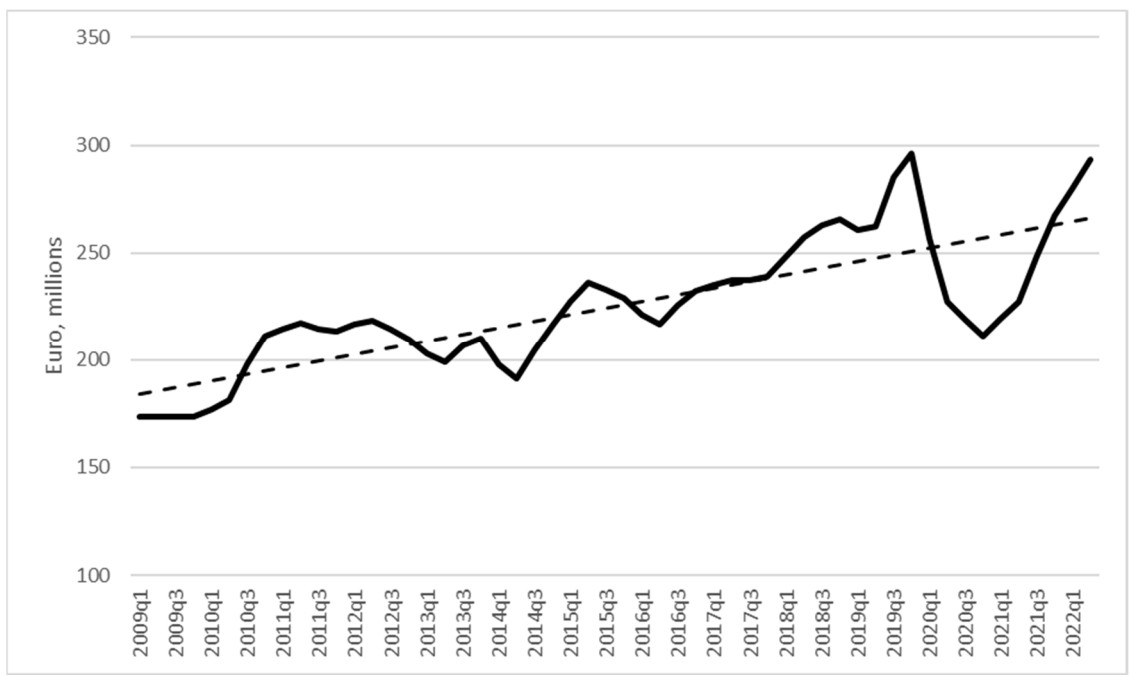

**Figure 1.** Ship management Revenue in Cyprus. The solid black line refers to the ship management companies' revenues in millions of euros, while the dashed line is the relevant trendline.

### 5. Estimation Results

Figure 2 shows the impulse response functions following the estimation of our Bayesian VAR model. As suggested in the literature, a shock in oil prices has the expected beneficial effects on stocks and the BDI index, while, as Michail and Melas (2020b) also find, higher oil prices tend to have a negative effect on BDTI and BCTI on account of the higher value of the cargo and the negative repercussions this will have on costs. As expected, revenues have a positive reaction to an oil price shock, suggesting that higher oil prices can potentially imply more demand, and most importantly, that ship management revenues are not static and fluctuate depending on the level of the freight rates.

Stock prices, as a proxy for the global macroeconomic environment, pose the largest source of change for ship management revenues. In particular, a 1% change in stock prices implies a 0.6% increase in ship management revenues, in contrast to around 0.1% following a 1% increase in oil prices. On the other hand, the change in BDI, BDTI, and BCTI is more evident in the longer run, given that only after 4–5 quarters do freight rates cause an increase in the indices. Given the expected delays in the shipping market before a shock is fully integrated (Michail and Melas 2023), this result is not out of the ordinary.

An interesting point for the researcher deals with the delay and freight rate absorption of the stock market shock and how it moves on to affect ship management revenues. A potential answer to that question lies in the anticipation and sentiment effects, which, as Melas and Michail (2021), have a strong impact on prices. As such, with higher sentiment about the future path of the world economy, ship management companies likely see this as an opportunity to expand their profit margins and thus request higher fees. While this is one potential explanation, we note that other factors, such as built-in terms in contracts, may also play a role.

As expected, the BDI and BDTI freight rates have a positive effect on revenues. However, this is not the case for BCTI, something that is perhaps attributed to the types of vessels under management. Of the two, the BDI has the largest impact on ship management revenues, standing at around 0.04% per 1% shock. The BDTI effect is lower at around 0.03%. Both results support the view that higher freight rates positively affect revenues, as was expected.

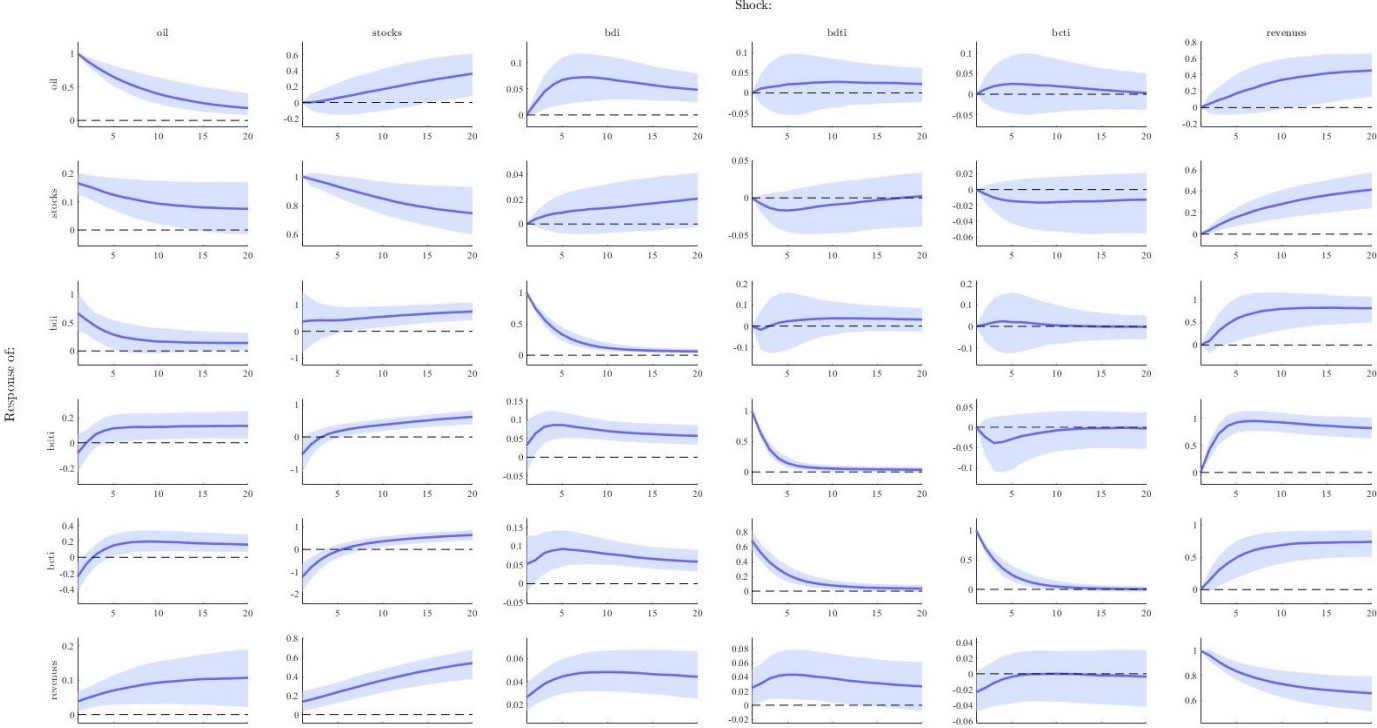

**Figure 2.** Impulse response functions. The solid blue line refers to the impulse response of the relevant variable to a shock, while the shaded area is the 68% confidence interval. For example, the first chart in the second line refers to the response of the stock market following a shock in oil prices.

As a general conclusion from Figure 2, it appears that the demand side has a strong effect on ship management revenues, with stock prices, used as a proxy for global macroeconomic conditions, having the largest impact on them. As suggested, this implies a sentiment effect, meaning that as stock prices rise, expectations of higher future gains make agents discount them to the present and raise ship management fees.

Moving to Figure 3, the main conclusions remain the same; however, when we add interest rates, their effect on freight rates and ship management revenues does not appear to be significant. This result adds value to the already existing bibliography since previous papers have found that shipping companies mitigate potential interest rate risks through hedging strategies (Mohanty et al. 2021; El-Masry et al. 2010). Naturally, while interest rates do not appear to have a direct effect, they can have an indirect impact via their influence on the stock market as the recent developments over 2022 have demonstrated (Eldomiaty et al. 2020).

Overall, the results suggest that the main macroeconomic factors do have a strong influence on ship management revenues. This can be important for both the companies as well as the local economies that depend on the well-being of the sector as they tend to be major players in the local labor markets. Furthermore, the implications include the fact that institutions that aim to evaluate the potential impact of adverse effects for forecasting purposes should also be able to properly forecast future ship management revenues given the importance of the sector. This, depending on the situation, can potentially have strong effects on a country's economic forecasts.

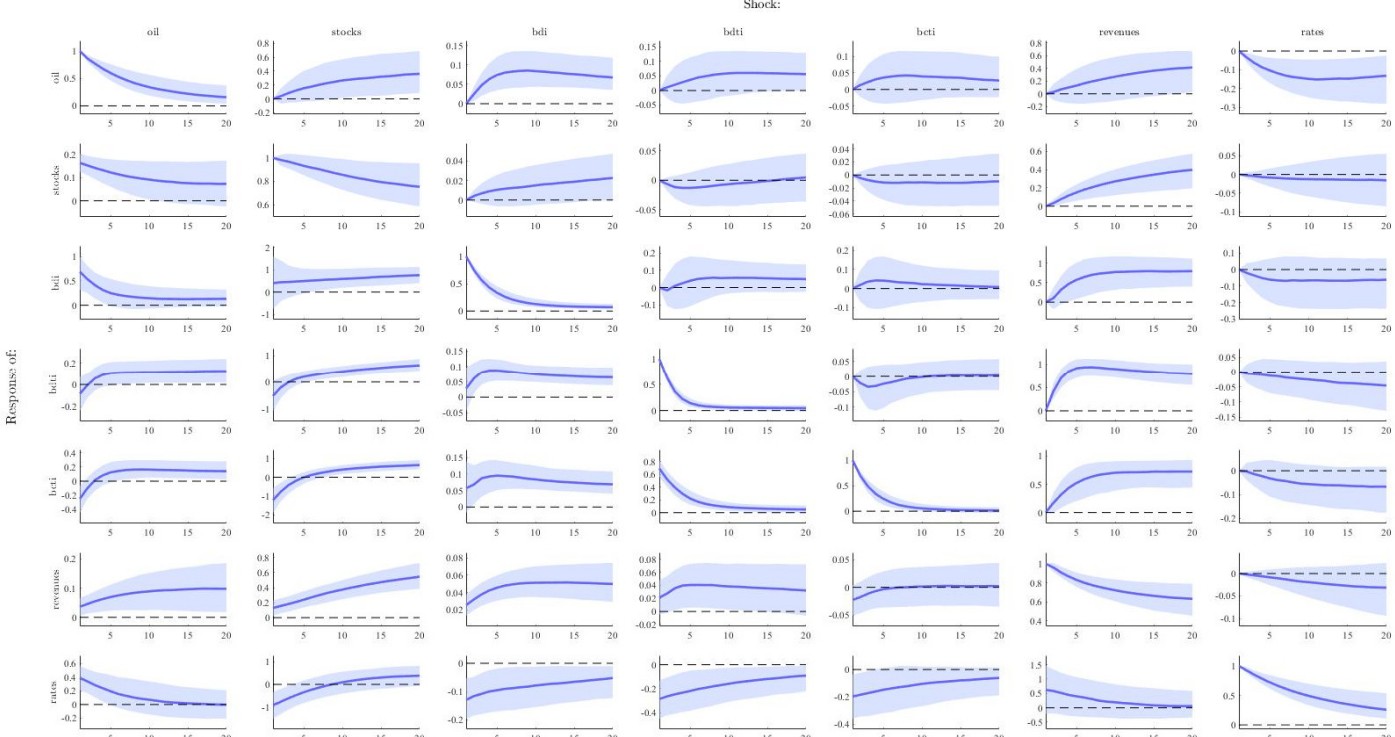

**Figure 3.** Impulse response functions (with interest rates). The solid blue line refers to the impulse response of the relevant variable to a shock, while the shaded area is the 68% confidence interval. For example, the first chart in the last line refers to the response of oil prices following a shock in interest rates.

## 6. Conclusions

We shed light on one of the most under-researched fields in the maritime industry, namely on the macroeconomic determinants of ship management companies. Despite their importance, little research has been conducted on the topic primarily due to the scarcity of the available information. In this paper, using a novel dataset from the Central Bank of Cyprus, we explored the interrelations between the freight rates of the vessels under management, interest rates, the stock market, Brent oil prices, and ship management revenues.

Our study has multiple findings. Initially, we found that both the dry bulk market and the dirty tanker market had a positive relationship with the revenues of the shipping companies. On the contrary, the clean tanker market did not seem to have an impact on ship management revenues. Thus, we can assume that the ship owners of clean tankers do not rely on third management services for their operations given the specific trade between distilled oil and consumers.

Additionally, we found a positive relationship between the stock market and the revenues of shipping companies. This result is especially interesting as it suggests that a general improvement in the macroeconomic outlook does not only affect markets but also boosts demand for transport, perhaps also by improving the overall sentiment in the industry (Michail and Melas 2021). Finally, we found that when we include interest rates in the estimation, these did not have an important effect on revenues. This can be attributed to the use of hedging strategies by companies.

Given the importance of this sector for many economies, with ship management revenues accounting for around 4.6% of the Cyprus GDP, changes in them can have a strong effect on a country. As such, the implications of our study include the ability to properly forecast future revenues as well as evaluate the potential impact of adverse effects, such as the recent pandemic or the effect of the interest rate-driven drops in the stock

market. This, depending on the situation, can potentially have strong effects on a country's economic performance.

Our research has, of course, its limitations, namely that we can only examine the results of the ship management companies that are located in Cyprus. As such, future work could be addressed for other countries that have ship management hubs, which would assist in providing further insights to this important sector.

**Author Contributions:** Conceptualization, K.D.M. and N.A.M.; methodology, N.A.M. and K.G.L.; formal analysis, N.A.M.; resources, N.A.M. and K.G.L.; data curation, K.G.L.; writing—original draft preparation, K.D.M. and N.A.M.; writing—review and editing, K.D.M. and N.A.M. All authors have read and agreed to the published version of the manuscript.

**Funding:** This research received no external funding.

**Informed Consent Statement:** Not applicable.

**Data Availability Statement:** Data is available upon request.

**Conflicts of Interest:** The authors declare no conflict of interest.

## Notes

[1] We also run the model with the use of additional lags in an effort to strengthen our findings. However the additional lags (i.e., 2, 3, and 4) do not appear to significantly change the responses. The IRFs are available upon request.

[2] While data from the CBC Ship Management Survey only exist at a semi-annual basis, we have interpolated them via a cubic spline to match the quarterly frequency of the rest of the data.

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
