# Peer review of "Determinants of Ship Management Revenues: The Case of Cyprus"

_economies, doi:10.3390/economies11070184_

Round 1

Reviewer 1 Report

This is a very interesting subject. The discussion is lean and cohesive. The methodology is properly utilized and the caveats are clearly stated.

The results are also comprehensive.  However, I have some minor suggestions / comments that might be useful for the author(s) in order to explore and potentially elaborate further in their discussion:

-I believe that the author(s) could also try additional model runs with lag >1; this might help with finding additional clues on the discussion with respect the differences (and different effects) in the results between BCTI and BDTI.
-I would suggest further research in commodity and and transportation market interactions recent bibliography as well as Park et al (2019) on  the role of maritime, land, and air transportation in economic growth.
-Dependent on the average fleet composition, there might be merit in replacing the BDI with BSI / BHSI / BPI.
-Enhancement of the discussion with respect to the theoretical differences between ship-owning companies (and their priorities) and ship management companies would also benefit the paper.
-Elements of the discussion and conclusions could be strengthened (or put under scrutiny) with Granger-causality tests in selected pairs.
-Some basic facts / statistics of the 200 ship mgt companies would also benefit the discussion (eg ship types, cargo types and other), and potentially provide additional clues. Commodities can impose varying effects.
-The non-significant effect of interest rates is not very surprising. There are similar results in recent bibliography.

We hope these comments help the author(s) enhancing the value of this very interesting paper.

Author Response

Please see the file attached.

Reviewer 2 Report

1) Variables (data) explicitely used in the estimated equation are poorely described. For instace, which stock index was taken, etc.

2) Is the selection of macroeconomic variables consistent with the fact that ship registration in Cyrpus has nothing to do with real opeartaions for paricular countries (as already stated in the paper)?

3) Diagnostic of the model is missing

Author Response

Reviewer 2

1) Variables (data) explicitly used in the estimated equation are poorly described. For instance, which stock index was taken, etc.

We would like to note that we already include variable specifications in the manuscript (page 12 - highlighted). For your ease, we repeat this here: we use Brent crude oil prices since they represent the main vessels expenses and the Wilshire 5000 total market full cap index because it captures the macroeconomic environment. In addition, the US Effective Federal Funds Rate (EFFR) is employed as a proxy for the policy rates and allows us to account for the potential spillovers from monetary policy and financing conditions. Since the US is the largest importer in the world, the US policy rate tend to have a global influence.

2) Is the selection of macroeconomic variables consistent with the fact that ship registration in Cyprus has nothing to do with real operations for particular countries (as already stated in the paper)?

We would like to thank the reviewer for this comment. In our paper, we do not examine any ship registration that takes place in Cyprus per se, but the income that ship management companies in the country generate. As such, given that these companies operate at a global level, we have included global factors to account for the variation in their revenue (see also our answer to your previous comment). This means that we take into account the fact that ship management revenue in Cyprus is of a global and not of a local nature.

3) Diagnostic of the model is missing

We are unclear as to what precisely the reviewer would like to see here. In the paper, we note that, for the VAR specification, standard hyperparameter values have been used, i.e., a 0.8 auto-regressive coefficient, tightness of 0.1, cross-variable weighting of 0.5, lag decay of 1 and 100 for the exogenous variable tightness. The variables have lag length of 1 and follow a Cholesky identification. All roots of the characteristic polynomial lie within the unit circle, making the VAR estimates statistically stable. R-squared values are high in all the relevant VAR equations. Please let us know if you would like to see any other diagnostics.

Round 2

Reviewer 2 Report

My comments were satisfactory answered, therefore I can suggest acceptance.